# Systematic Discovery of FBXW7-Binding Phosphodegrons Highlights Mitogen-Activated Protein Kinases as Important Regulators of Intracellular Protein Levels

**DOI:** 10.3390/ijms23063320

**Published:** 2022-03-19

**Authors:** Neha Singh, András Zeke, Attila Reményi

**Affiliations:** Biomolecular Interactions Research Group, Research Center for Natural Sciences, Institute of Organic Chemistry, H-1117 Budapest, Hungary; neha.singh@ttk.hu (N.S.); zeke.andras@ttk.hu (A.Z.)

**Keywords:** FBXW7/FBW7/CDC4, ubiquitin ligase, MAP kinase, ERK, JNK, p38, docking motif, phosphodegron, cancer

## Abstract

A FBXW7 is an F-box E3 ubiquitin-ligase affecting cell growth by controlling protein degradation. Mechanistically, its effect on its substrates depends on the phosphorylation of degron motifs, but the abundance of these phosphodegrons has not been systematically explored. We used a ratiometric protein degradation assay geared towards the identification of FBXW7-binding degron motifs phosphorylated by mitogen-activated protein kinases (MAPKs). Most of the known FBXW7 targets are localized in the nucleus and function as transcription factors. Here, in addition to more transcription affecting factors (ETV5, KLF4, SP5, JAZF1, and ZMIZ1 CAMTA2), we identified phosphodegrons located in proteins involved in chromatin regulation (ARID4B, KMT2E, KMT2D, and KAT6B) or cytoskeletal regulation (MAP2, Myozenin-2, SMTL2, and AKAP11), and some other proteins with miscellaneous functions (EIF4G3, CDT1, and CCAR2). We show that the protein level of full-length ARID4B, ETV5, JAZF1, and ZMIZ1 are affected by different MAPKs since their FBXW7-mediated degradation was diminished in the presence of MAPK-specific inhibitors. Our results suggest that MAPK and FBXW7 partnership plays an important cellular role by directly affecting the level of key regulatory proteins. The data also suggest that the p38α-controlled phosphodegron in JAZF1 may be responsible for the pathological regulation of the cancer-related JAZF1-SUZ12 fusion construct implicated in endometrial stromal sarcoma.

## 1. Introduction

The level of proteins in cells is the result of a dynamic equilibrium between production and degradation. Proteins are produced by translation and are mostly eliminated by proteosomal degradation. Proteins are destined to be degraded by ubiquitination and the process is mediated by ubiquitin ligases. The overall system of ubiquitination and proteasomal degradation involves several ubiquitinating enzymes (E1 and E2), but the decisive role is played by E3 ubiquitin ligases that are responsible for targeting and selecting proteins for proteosomal degradation.

Humans have an estimated number of ~600 E3 ligases classified into four families [1]. They are often multi-protein complexes: Skp1-Cullin-F-box (SCF) consists of four proteins, where three are invariant, but the F-box varies and is responsible for substrate specificity. There are ~70 human known F-box proteins (FBX) that recognize their clients through different domains and 12 FBXW proteins contain a WD40 domain important for substrate binding [2]. FBXW7, also known as Cdc4, binds the degron Thr-Pro-Pro-Xaa-Ser sequence in which the Thr and Ser residues are phosphorylated (and Xaa represents any amino acid) [3]. Intriguingly, the *FBXW7* gene is often disrupted in different types of cancer or the WD40 domain is affected by missense point mutations [4,5]. Conversely, phosphodegron motifs recognized by FBXW7 are often affected by cancer-associated mutations, particularly in FBXW7 controlled tumor suppressors [6].

FBXW7 is known to be a key player in many solid tumors and leukemias [7]. It is among the most highly mutated genes in endometrial, colorectal, and cervical cancers according to COSMIC data [8,9]. It is known that the *FBXW7* gene is frequently mutated in acute lymphoblastic T-cell leukemias and the degron motif containing region of NOTCH1—a key substrate of this ubiquitin ligase—is also mutated as an alternative, if *FBXW7* remains intact [10]. Moreover, the degron in KLF5 is also mutated as an alternative to FBXW7 ablation in colorectal cancer [11]. In most solid tumors, however, there is barely any mutation of degrons belonging to a known substrate of FBXW7, which may suggest that there could be a multitude of substrates that are simultaneously responsible for carcinogenicity. For cervical and endometrial cancer, we have no hints on what protein(s) might transduce the effects of FBXW7 ablation. Therefore, we decided to adopt a “bottom-up” approach and to predict substrates directly from protein sequences. Our intention was to identify those substrates that may have been missed by former studies due to their low abundance or highly tissue- or cell-cycle dependent expression pattern.

FBXW7 controlled phosphodegron motifs are normally located in disordered parts of proteins where first they need to be phosphorylated by protein kinases, and then they bind to the WD40 domain as bisphosphorylated short motifs (~10 amino acids). How can such short regions specifically control protein degradation? The crystal structure of FBXW7-Skp1 complex with a bisphosphorylated degron motif from cyclin E1 (CCNE1) provided the structural basis of this phosphorylation dependent interaction [12]. Moreover, several studies highlighted that GSK3, a constitutively active kinase, often phosphorylates FBXW7 binding degrons [13,14,15] and cyclin-dependent kinases (CDKs) are also known to phosphorylate FBXW7 degrons in some proteins whose level changes in the cell cycle [16,17,18,19].

Although the phosphorylation of FBXW7 degrons as a requirement for FBXW7 WD40 binding is well-established, the protein kinases that activate FBXW7 degrons are often not known. Mitogen-activated protein kinases (MAPK), which are activated by extracellular stimuli, had formally been implicated in FBXW7 mediated protein level control. For example, FBXW7 was shown to bind to phosphodegrons in c-Jun to antagonize apoptotic c-Jun N-terminal kinase (JNK) signaling in neurons [20], albeit another study later found that it is GSK3 that plays the decisive role in this process with another unidentified kinase together [21]. GSK3 recognizes substrates “primed” by phosphorylation so that phosphodegrons in general would be formed by the action of a priming kinase and GSK3: GSK3 and p38 for example both play an important role in FBXW7 dependent degradation of the peroxisome proliferator-activated receptor gamma coactivator-1α (PGC-1) protein and a simple consensus motif drawn up based on known FBXW7 phosphodegron motifs—[TS]PPx[TS]P, where S is serine, T is threonine, P is proline, and x is any amino acid—is compatible with the phosphorylation target site (S/TP) of mitogen-activated protein kinases (MAPKs) [22].

MAPKs affect protein function by targeting phosphoswitch regions in their substrates [23]. These short regions change their biochemical properties upon phosphorylation and may control functionally diverse processes (e.g., enzyme activity, protein localization, and gene expression) [24]. MAPKs phosphorylate their substrates on S/TP target sites far more efficiently if the substrate harbored a docking motif (e.g., D(ocking)-motif or FXFP motif) [25]. These short, disordered protein regions bind to the MAPK docking groove which is distinct from the kinase’s classical substrate binding pocket [26,27]. Co-occurrence of a MAPK binding docking motif and a MAPK phosphorylation compatible FBXW7 phosphodegron in one protein may hint at a cooperation between these two key post-translational mechanisms.

In the present study the phosphorylation dependent FBXW7 sequence motif has been refined through an iterative process involving proteome level sequence pattern searches and experimental testing of putative motifs. Out of 55 putative motifs tested, we experimentally validated phosphodegron motifs from 18 different proteins (19 novel motifs). Degradation of some of these new FBXW7 clients was inhibited by MAPK-specific inhibitors suggesting that MAPK and FBXW7 partnership affects their turnover rate in the cell. Finally, we demonstrate that the p38α-controlled phosphodegron in JAZF1 may be responsible for the pathological regulation of the cancer-related JAZF1-SUZ12 fusion construct implicated in the early stages of endometrial stromal sarcoma (ESS).

## 2. Results

### 2.1. FBXW7 Binding Motif Identification by a Ratiometric Protein Degradation Assay

In order to be able to test putative phosphodegron motifs, a ratiometric protein level monitoring method was adapted to monitor MAPK controlled FBXW7 mediated protein degradation [28]. Briefly, intact DsRed and an EGFP-chimera were expressed in HEK293T cells in equal amounts from an expression plasmid, giving an EGFP/DsRed fluorescence ratio close to one under non-degrading conditions. EGFP was fused to a probe comprised of a putative degron sequence and a known MAPK binding docking motif. EGFP signal is expected to diminish faster than DsRed in cells with more FBXW7 if the probe responded to degradation. Therefore, low EGFP/DsRed ratio indicates high degradation rate for the EGFP chimera suggesting that the tested degron motif is functional (Figure 1A).

The assay was first validated on the cyclinE degron motif which is known to undergo cyclin-dependent kinase mediated phosphorylation [29]. This sequence was modified and made compatible to MAPK mediated phosphorylation by introducing a proline after the second phosphorylation site (CycECO: cyclinE C-terminal optimized degron). We first monitored FBXW7 mediated proteosomal degradation of the CycECO probe containing a V5-tag in HEK293T by western blots. Cells, untreated or transfected with an FBXW7 expression plasmid, were treated with a proteosome inhibitor (MG132) or with a translation inhibitor (cycloheximide): the protein level of the CycECO-V5 probe was diminished in cells transfected with an FBXW7 expression plasmid and MG132 counteracted the effect of FBXW7 expression, moreover, blocking translation by cycloheximide treatment caused a more rapid drop in the V5 western blot signal in FBXW7 transfected cells (Figure 1B). These results suggested that the designed degradation probe is affected by FBXW7 as expected. The protein level of the probe was similarly affected when monitored in the ratiometric EGFP/DsRed assay (Figure 1C). Further tests with other probes containing well-known FBXW7 degron motifs (e.g., MYC, PGC1A, and SOX9) also confirmed that EGFP/DsRed signal ratio is a suitable proxy for degron motif mediated degradation because (1) EGFP/DsRed was dependent on the presence of elevated amounts of FBXW7, (2) its decrease was counteracted by proteosome inhibitor treatment (MG132), and (3) serine to alanine mutation in the phosphorylation target sequence rendered the EGFP chimera unresponsive to FBXW7 (Figure 1D).

Next, we tested the impact of the MAPK docking motif in the designed degradation probes. All probes contained a known MAPK binding short sequence from a p38/ERK binding protein, AAKG2 [23]. Disruption of endogenous MAPK recruitment by mutating key residues pivotal to binding in the MYC, PGC1A, and SOX9 degron motif containing constructs diminished degradation of the probe (Figure 1E). FBXW7 mediated degradation of the optimized cyclin E1 motif was unaffected, suggesting that this motif receives phosphorylation that is independent from MAPKs, possibly from cyclin-dependent kinases (CDKs). The role of MAPKs in the phosphorylation of different degradation probes was then confirmed by using MAPK selective inhibitors (Figure 1F). The inhibition exerted by these MAPK-specific inhibitors on FBXW7 mediated degradation was only partial. This may be because the degradation experiments were performed under low MAPK activity conditions, other kinases may also be active on the degradation probes, or mono- or double-phosphorylation of the phosphodegron motif may have somewhat distinct effects. For the final validation step of the assay, 10 additional reported FBXW7 binding motifs were tested. All EGFP chimera, apart from MCL1, showed increased rate of degradation upon FBXW7 expression (Figure 1G).

### 2.2. Discovery of New FBWX7 Phosphodegrons in the Human Proteome

The assay was next used to test the importance of key positions in the CycEco phosphodegron (Figure 2A). The first threonine (P0) can be replaced by serine, although this makes the degron weaker. In contrast, replacement of the phosphorylatable residue to phospho-mimicking aspartate or glutamate rendered the motif to be nonfunctional. Mutating the P + 2 proline to alanine makes the degron also nonfunctional, however, exchanging it to glycine is tolerated, albeit the motif became weaker. The proline in P + 2 can be moved to P + 3 position. In addition, adding one proline or shortening the intervening region between the two phosphorylatable sites by one residue is also tolerated. The second phosphorylatable site (P + 4) cannot be replaced to alanine. These results are in line with the observation that strong FBXW7 motifs need to possess a core TPP-like element with appropriate flank regions on both sides. Motifs that violate these rules in some positions (e.g., P + 2 or P + 4) may mediate only weak degradation. However, these studies also demonstrate substantial flexibility in the sequence of functional degrons, given that other positions were kept optimal or nearly optimal. As the φ1-φ2-Ser/Thr-Pro-Pro-X-Ser/Thr-Pro sequence pattern was the most common among known motifs (where φ1 and φ2 are any hydrophobic and small hydrophobic residues, respectively), we started searching for similar sequences as well as for their variants that were close enough to this original motif.

Based on the results of the degradation assay and the data obtained from the mutational analysis with the CycEco motif, we looked for a way to identify putative FBXW7 degron motifs in the human proteome. We used a multi-step approach where sequence pattern-based prediction was followed by experimental testing of motifs in the protein degradation assay. Starting out with a very promiscuous motif observed for yeast Cdc4 substrates, we progressively restricted the sequence based on the known and tested motifs as well as the published structures [30]. SlimSearch4 was used as the initial filtering algorithm to collect sequence pattern matches in the disordered part of the human proteome [31] (Figure 2B). Sequence patterns were updated in four cycles based on the experimental results obtained in the former cycle (Appendix A). More than 50 motifs were tested in the assay and close to one-third of these turned out to be positive. The identified 19 novel phosphodegron motifs showed comparable degradation signal to those motifs that had been found positive and functional earlier (Figure 2C).

The availability of a high number of positives (19 confirmed positives) and tested negatives (37 confirmed negatives) allowed a substantial refinement of the functional FBXW7 motif based phosphodegron sequence space (Figure 2D). It appears that the FBXW7 phosphodegron motif has clear sequence preference in several key positions, but it is also malleable to a certain degree. For example, the set of negatives implied that P−1 and P−2 do not tolerate every residue and are important restricted positions. Similarly to data obtained with the CycEco motif, the analysis on diverse putative motifs also suggests that serine can be tolerated in P0 and the second phosphorylation site (P + 4) may also be Glu/Asp instead of a phosphorylatable residue [18,32,33]. In summary, motif positions may be classified as less/not restricted, restricted or strictly restricted in the new FBXW7 phosphodegron logo (Figure 2D). These observations are well in-line with the structure of known FBXW7-substrate complexes and with the coordination of the central substrate region (Figure 2E).

The sequence rules captured in the refined sequence logo can be rationalized based on the several requirements that a phosphodegron motif has to comply so that to be functional. In addition to efficient binding to FBXW7, requiring that the bisphosphorylated motif must be compatible with the WD40 domain surface, the motif sequence also needs to be compatible with the substrate binding pocket of the activating protein kinase. Furthermore, the phosphorylated protein region binds to a shallow protein–protein interaction surface with two charged contacts on the WD40 domain, which may also confer some restraints on positions in the restricted region. Another important observation is that many FBXW7 motifs do not confer to a single consensus, instead they are “close enough”: implying a single relaxation from a theoretically optimal motif in most cases (where CycECO may be close to an optimum). Based on this new sequence information, we set up seven different FBXW7 degron classes: optimal, P − 2 relaxed, P − 1 relaxed, P0 relaxed, P + 4 D/E, P + 4 nonphospho, and P + 2/P + 3 exchanged, which deviate only at one position from the optimal motif pattern, apart from the last category that has a “TPxP” core. Our set of positives and negatives suggests that a single relaxation form the optimum (defined by the regular expression [^R][PLIVMFY][PLIVM]TPP[^RG][ST]) is well tolerated as long as it is picked from a list of permitted amino acids for each position. On the other hand, multiple relaxations are deleterious and mostly tested as negatives or gave weak binders (e.g., JUN). Therefore, we established a new position–relaxation model which takes into account these observed differences (Figure 3A). For example, an aspartate or glutamate in P + 4 (P + 4 D/E class) is tolerated instead of a more optimal phosphorylated serine/threonine, and this position may even tolerate arbitrary amino acids apart from positively charged arginine or lysine (P + 4 nonphospho class) in motifs that are absolutely optimal elsewhere. At P0, in addition to the optimal phosphorylatable threonine, serine may also occur. Because of the hollow topology of the WD40 domain interfacing with the P − 1 or P − 2 residues, it appears that a greater variety of amino acids may also be tolerated at these positions.

When we used this more relaxed motif set, we could correctly identify 28 known motifs from the “TPP.[STED]”-like core pattern (or TPxP[STED]) and only missed three (JUN, SNX8, and DISC1), putting the sensitivity of the approach to be quite high for positives (>80%; see Appendix A). Similarly, this method correctly excluded 31 negatives out of 37 FBXW7 motifs tested (see Appendix A), giving a specificity > 80% for identifying negatives (Appendix A).

Next, we characterized some of the newly identified phosphodegron motifs further. A fluorescence polarization based quantitative in vitro binding assay confirmed that all bisphosphorylated peptides representing different motif classes (e.g., optimal: ETV5 and KLF4; P − 1 or P − 2 relaxed: KAT6B, KLF4, ARID4B, JAZF1, and ZMIZ1) bound to purified FBXW7 in vitro with low nanomolar binding affinity, similarly to the known CCNE1 motif (optimal class). In addition, the binding of a motif from the P + 4 nonphospho class (CCAR2) was also confirmed, albeit this peptide interacted only with low micromolar binding affinity (Figure 3B).

### 2.3. Characterization of New FBXW7 Binding Degrons in Full-Length Proteins

Next, we selected 6 of the 19 new phosphodegron motifs (see Figure 2C) and addressed how they would function in their full-length protein context. ETS translocation variant 5 (ETV5) and Krueppel-like factor 4 (KLF4) are transcription factors [34,35]. CDT1 is a DNA replication factor [36]. Zinc finger MIZ domain-containing protein 1 (ZMIZ1) is a transcriptional co-activator of some of the NOTCH1 target genes (e.g., MYC) [37]. Juxtaposed with another zinc finger protein 1 (JAZF1) and AT-rich interactive domain-containing protein 4B (ARID4B) are transcriptional corepressors [38,39]. The cDNA of full-length KLF4, ETV5, ARID4B, CDT1, ZMIZ1, and JAZF1 were cloned into the DsRed-EGFP-chimera plasmid. HEK293T cells were transfected with these constructs and protein levels were measured by the DsRed/EGFP ratiometric protein level monitoring method in control cells and in cells co-transfected with an intact (WT) or a phosphodegron-binding deficient version of FBXW7 (Arg Mut: three arginine residues in the WD40 domain indispensable for phosphopeptide binding were mutated to amino acids occurring in cancer-associated mutant versions of the protein: R465C; R479Q; R505L). All six constructs showed significant FBXW7-mediated degradation, which required an intact WD40-phosphodegron binding interface (Figure 4A). These results were also qualitatively confirmed by western blots (Figure 4B). Moreover, MG132 counteracted the effect of FBXW7, and protein levels showed a relative increase in the presence of this proteosomal inhibitor indicating that the observed protein level decrease in the presence of FBXW7 depends on proteosome mediated degradation (Figure 4C). Finally, when the two phosphorylatable residues (in P0 and P + 4) were replaced to alanines (TA Mut) FBXW7-mediated degradation was not observed, indicating that mutations of these phosphosites in the FBXW7 client proteins—similarly to the arginine mutations in the WD40 domain of FBXW7—are also deleterious to protein degradation as expected (Figure 4D). In conclusion, these experiments showed that the new phosphodegron motifs identified in the earlier described screen using artificial reporters are also functional in their full-length protein context and may govern the degradation of intact proteins with physiologically diverse functions.

### 2.4. Protein Kinase Mediated Phospho-Regulation of FBXW7 Binding Degrons

GSK3 was earlier shown to affect the degradation rate of many known FBXW7 client proteins [40]. In order to test if the degradation of the new full-length constructs can be affected by this constitutively active kinase, we used a GSK3-specific inhibitor (CHR99021) but found no effect of this compound in the ratiometric protein level monitoring assay [15]. In contrast to this finding, MAPK-specific inhibitors (ERK1/2—SCH772984, p38—SB202190, and JNK—JNK-IN-8) had a significant effect on FBXW7-mediated protein degradation in four cases out of six full-length targets tested [22,41,42]. Notably, the degradation of JAZF1 was lower in the presence of the p38-specific inhibitor, the JNK-specific inhibitor had a significant effect on ARID4B and ETV5 levels, while ZMIZ1 level was affected by all three MAPK inhibitors (Figure 5). The effects of MAPK inhibitors were modest in most cases, possibly because other kinases are also active on these phosphodegrons. Notwithstanding, the data suggest that certain MAPKs may specifically be involved in the protein level regulation of ZMIZ1, ARID4B, ETV5, and JAZF1 by regulating recognition by FBXW7.

### 2.5. p38-Controlled Regulation of an Endometrial Stromal Sarcoma Associated JAZF1 Fusion Protein

The in-frame fusion of the JAZF1 and SUZ12 genes, due to (7; 17) (p15; q21) chromosomal translocation or RNA trans-splicing under hypoxic condition, is present in most cases of endometrial stromal sarcomas [43,44]. Other similar JAZF1 fusions may happen with PHD finger protein 1 (PHF1), a polycomb group protein that binds to histone 3 trimethylated at lysine 36 and recruits the polycomb repressive complex 2 (PRC2) [45,46], or with BCORL1, a transcriptional co-repressor [47] (Figure 6A). These fusions involve JAZF1 (exon 1–3) with SUZ12 (exon 2–16), PHF1 (noncoding region before exon1 to maintain the frame) and BCORL1 (exon 5–12 or 7–12). It is currently unknown why these fusions are associated with uterine tumors, let alone with new pathogenic subgroups of ESS, but it is intriguing that two of these known JAZF1 fusions (with PHF1 or SUZ12) could directly alter PRC2 function. This regulatory complex contains three core proteins EZH2, EED, and SUZ12 and controls chromatin compaction and transcription repression through trimethylation of lysine 27 on histone 3 [48]. Since these fusions all involve the transfer of the N-terminal JAZF1 region (1–129 or 1–130) containing the FBXW7 binding phosphodegron motif (109-TPPVTP-114) to other regulatory proteins, we posited that protein levels may be differently affected compared to intact proteins. To this end, we generated the endometrial stromal sarcoma associated JAZF1-SUZ12 fusion and investigated how FBXW7 affects its protein level compared to intact JAZF1 and SUZ12. The protein level of JAZF1 was greatly decreased by FBXW7 when cells were co-transfected with the intact ubiquitin ligase but remained unchanged with the mutated WD40 domain containing construct (Arg Mut), and the level of SUZ12 was unaffected as expected. However, the degradation of the JAZF1-SUZ12 fusion greatly increased and showed an FBXW7 dependent pattern similar to full-length JAZF1 (Figure 6B).

The N-terminal region of JAZF1 contains a MAPK-binding D-motif that was shown to bind p38α: 77-KKKIQPKLSLTL-89 (where positively charged or hydrophobic amino acids binding into the negatively charged CD-groove or the hydrophobic pockets of the D-groove, respectively, are underlined) [23] (Figure 6C). In agreement to this, examination of the degradation of the JAZF1-SUZ12 fusion construct showed that its FBXW7-mediated increase could be counteracted by a p38-specific inhibitor (SB202190) (Figure 6D).

## 3. Discussion

FBXW7 degrons require phosphorylation in order to be recognized by the ubiquitin ligase. This phosphorylation is best promoted by proline-directed kinases since the structure of the WD40 domain binding surface interacting with the bisphosphorylated peptide fits best with a proline after the key phosphorylated residue (S/T-P target motif consensus at P0 and P + 1; see Figure 2E). Some degron motifs are autonomous because they contain a good kinase target site at P + 4 or they have a phospho-mimicking aspartate or glutamate at this position. The FBXW7 binding degrons in SOX9 and SOX10 were shown to be phosphorylated by the constitutively active GSK3 at both P0 and P + 4 [49,50]. Similarly, casein kinase II (CKII) is sufficient alone to recruit β-TrCP, another ubiquitin ligase from the SCF family, to its degron motif in the E2 ubiquitin conjugating UBC3B protein [51]. However, a larger portion of these degrons tend to depend on conditionally active kinases. For example, CDK1, a cell-cycle controlled enzyme, drives FBXW7 mediated degradation of the GATA2 transcription factor, since the latter contains the [ST]..[KR] motif for cyclin-dependent kinases [18].

Naturally, FBXW7 phosphodegrons might require more than one kinase: in addition to constitutively active kinases (such as GSK3 or CKII), dual phosphorylation of less autonomous degrons would depend on regulated kinases such as CDKs, affected by the cell cycle, or on MAPKs, affected by extracellular stimuli. Thus, efficient phosphorylation of some degrons in vivo, in the presence of counteracting phosphatases, may depend on the dual action of a master kinase and a so-called slave kinase (e.g., GSK3 or CKII) where the recruitment of the latter would depend on its priming site phosphorylation by the former [52,53]. Moreover, the recruitment of the regulated master kinase may also depend on additional kinase recruitment motifs outside of the degron motif and could be governed by different types of docking motifs for CDKs and/or MAPKs in particular [23,54]. Alternatively, some well-positioned CDK or MAPK docking motifs may enable efficient phosphorylation of the degron by these regulated kinases alone. For example, the centrosomal phosphoprotein NDE1 is known to be phosphorylated by CDK5 alone [17], while CCNE is phosphorylated by CDK2 and the primed degron is further processed by GSK3 [55]. Other examples include the PGC1α phosphodegron which is primed by the p38 MAPK at P + 4 position and GSK3 further phosphorylates it at P0 [22], or the c-Myc degron similarly controlled by ERK1/2 and GSK3 together [52]. Furthermore, it appears that the same phosphodegron in one protein can be phosphorylated by different kinases depending on the cellular and physiological context. For example, HSF1 is phosphorylated by CKII in Huntington’s disease affected brain cells but the same degron motif is primed by ERK1/2 and is further phosphorylated by GSK3 in cancerous cells [56,57]. Our data suggest that JAZF1 is controlled by p38 and the protein turnover of ARID4B and ETV5 is affected by JNK, while ZMIZ1 appears to be affected by all three MAPKs. We found that GSK3 does not play an important role in the phosphorylation of the degrons from the four full-length proteins that we analyzed. This may be because our experimental pipeline for FBXW7 degron motif discovery was intentionally biased towards MAPK-based regulation: the degron motif chimera constructs contained a MAPK selective recruitment sequence (see Figure 1) and the basal level of MAPK activity in HEK293T cells cultured in complete media drove their phosphorylation dependent degradation on its own, independently from other kinases. In agreement to this, all new motifs tested positive in our degradation assay had MAPK compatible S/TP sites (see Appendix A).

Most of the known FBXW7 targets are localized in the nucleus and function as transcription factors. In this study, in addition to more transcription factors (ETV5, KLF4, SP5, JAZF1, and ZMIZ1 CAMTA2), we identified phosphodegrons located in proteins involved in chromatin regulation (ARID4B, KMT2E, KMT2D, and KAT6B), cytoskeletal regulation (MAP2, Myozenin-2, SMTL2, and AKAP11), and some other proteins with miscellaneous functions (EIF4G3, CDT1, and CCAR2) (Figure 7A). The Kruppel-like factor family consists of 17 transcription factors (KLF 1-17), and 5 of them contain similar phosphodegrons. The degron in KLF4 is found only in one of its known isoforms (isoform 1; 513aa), which is 34 amino acids longer [58], due to the retention of an intron because of alternative splicing, than its more widely known version (isoform 2; 479 aa). Interestingly, this short extra region harbors an FBXW7 binding phosphodegron (Figure 7B). Another interesting new example is the phosphodegron motif in ZMIZ1 that can function as a transcriptional co-activator together with NOTCH1 [37]. All four motifs (JAZF1, ETV5, ARID4B, and ZMIZ1) found to be affected by MAPK inhibitors are evolutionary conserved in vertebrates. Furthermore, an evolutionarily conserved degron motif was identified in the histone acetyltransferase KAT6B, while this motif is absent in its KAT6A paralog (Figure 7C). The sequence analysis of SMTL2, a muscle differentiation affecting protein in mammals, may reveal an interesting example of co-evolution between FBXW7 and MAPK based regulation. SMTL2 was reported to have a JNK-binding D-motif directing the phosphorylation of this protein [59]. Interestingly, we showed earlier that this MAPK-binding site emerged by gradual point mutations and is found only in placental mammals [23], but more importantly a similar analysis showed that the FBXW7 binding phosphodegron may have also emerged, and appears to co-occur with the MAPK binding docking motif among SMTL2 orthologs, albeit the D-motif seems to have somewhat preceded the degron motif in mammals (Figure 7D).

FBXW7 phosphodegrons are biochemically interesting examples of domain-linear motif type protein–protein binding. Apart from that the phosphorylated motif needs to be able to bind into the shallow WD40 domain binding slot, the motif needs to be compatible with protein kinases that phosphorylate them. Because of the specific topology of the phosphopeptide binding slot on the WD40 domain, extracellular signal or stress-regulated proline-directed kinases such as MAPKs could potentially play an important role in this. In this present study we identified 19 motifs located in different proteins that bind as bisphosphorylated motifs to FBXW7 with high affinity and showed that MAPKs can indeed affect the level of these proteins in cells. We posit that MAPKs may partner up with FBXW7 to set the level of ubiquitin ligase substrates depending on MAPK signaling pathway activity. Conversely, pathological up-regulation of MAPK pathways, which is a general hallmark in cancers, may contribute to the disease in its early stages before FBXW7 goes through some loss of function mutations. It appears that none of the new validated degron motifs have increased cancer-associated somatic mutation rate (based on COSMIC data analysis, not shown), possibly because their loss-of-function effect is individually not strong enough, albeit the degrons located in the MYC proto-oncogene and KLF5 were found to be a hot-spot in B-cell lymphomas or colorectal cancer, respectively [60,61]. Conversely, when the degron motif is taken out from its natural regulatory context, for example in the case of the JAZF1-SUZ12 gene fusion, the gain-of-function effect of the degron could be detrimental under specific regulatory conditions, since it will alter the degradation rate of otherwise stable proteins. Based on our experiments using artificial constructs, to our knowledge, our study is the first that suggests that natural JAZF1 level could be regulated by the FBXW7 ubiquitin ligase in a MAP kinase dependent fashion. Moreover, we suggest that JAZF1-SUZ12 fusion protein level could be pathologically affected due to the illicit rearrangement between the protein level controlling module of JAZF1 and a polycomb group protein (JJAZ1/SUZ12). Hypoxia induces p38 activation and the endometrial stroma is naturally exposed to hypoxic conditions during the female menstrual cycle. Speculatively, the pathological effects of the JAZF1-SUZ12 fusion may be exacerbated, or possibly driven because of p38-mediated phosphorylation of the JAZF1 phosphodegron motif. The latter could drive the pathological ubiquitylation of the polycomb repressive complex 2 (PRC2), ultimately leading to less H3K27 methylation and, thus, to global chromatin de-repression [48,62]. Because of this and based on the outcome of our study regarding the involvement of p38 in JAZF1-SUZ12 fusion protein level regulation, we posit that p38 inhibition may ameliorate symptoms associated with PRC2 complex dysfunction observed in endometrial stromal sarcoma (ESS).

## 4. Materials and Methods

### 4.1. Phosphodegron Motif Selection

We started out from an intentionally general consensus motif of FBXW7 degrons, inspired by the liberal [TS]P..[TS]P pattern defined for yeast Cdc4 substrates [63], and scanned the human disordered part of the proteome using SliMSearch4 (the disorder score cut-off was 0.4 and flank length was 5) [31]. In all, 22 motifs from this list were subjectively chosen and tested in the ratiometric degradation assay giving only 5 positives (see Appendix A). Based on the results of this first round, by taking information both from the identified positives and negatives, the consensus motif definition was narrowed to requiring a proline at P + 2, but motifs were allowed to have a phospho-mimicking residue (D or E) at P + 4 because it had been formerly reported that this second phosphorylatable residue may be replaced by aspartate or glutamate [3]. The experimental hit rate for this second cycle was also low: 5 motifs out of 21 were found to be positive. In the third round, hydrophobic amino acids were more strictly required at positions −1/−2 and P − 1/2/3 positions were further narrowed and P + 4 position was relaxed based on the identified negatives, which gave a higher hit rate (4 out of 6). This third round gave functional motifs at a good rate, but the used motif definition was very restrictive compared to other known motifs. Finally, in the fourth round, we gave up using an optimal consensus and established the positional relaxation model. In the last round, 5 out of the 6 tested motifs were found to be positive. Hits on this final, most predictive list are presented in Appendix A. All entries were filtered for motif locations by UniProt annotations, discarding all enzyme-inaccessible (e.g., extracellular) instances. Motifs that had any overlapping PFAM domain indicated were manually assessed using the AlphaFold EBI website. Out of these, only motifs that were highly accessible and located outside of confidently predicted folded domains were kept. We also added kinase consensus compatibility to every motif and the number of experiments where phosphorylation could be detected from PhosphoSitePlus [64].

### 4.2. Plasmids and Construct Design

The chimera degron module was ordered from IDT as gene Block (gBlock). It consisted of a coiled coil dimerization motif (GCN4: EELLSKNYHLENEVARLKK) to increase sensitivity by allowing multivalent binding, a MAPK binding docking motif (AAKG2: KKNASQKRRSLRVHIPDL), and the CycECO degron motif. The motifs were separated by Gly-Ser-Asn linkers and the gBlock was designed so that the motifs could be easily changed by using unique sets of restriction endonucleases. The gBlock was sub-cloned into the Gateway pENTR4 entry plasmid (Thermo Fisher, Waltham, MA, USA) using restriction enzymes (SalI and NotI). The pENTR4-chimera and the destination plasmid (MSCV-CMV-DsRed-IRES-EGFP-DEST from Addgene; 41941), [28]) were subjected to gateway cloning using LR clonase II enzyme (Invitrogen) to obtain the first DsRed-EGFP-chimera DNA construct containing the cycECO degron. Oligonucleotides corresponding to different degron sequences and docking motifs were ligated into the DsRed-EGFP-chimera DNA construct. The list of all the oligonucleotides used for cloning is provided in Appendix A. pcDNA3-FLAG-FBXW7α was a kind gift from Michael Pagano (New York University). The FBXW7 arginine mutant (Arg mutant: R465C; R479Q; R505L) was created using two-step PCR and cloned into the pcDNA3-FLAG plasmid. The cDNA of JAZF1, ETV5, SUZ12, and Skp1 were obtained from HEK293T cell cDNA pool by reverse transcription. Other full-length clones were obtained from plasmid DNA repositories; Addgene: CDT1 (117500), KLF4 (17227); DNASU: ARID4B (HsCD00829712); Biocat ZMIZ1 (BioCat, #BC172361-TOH6003-GVO-TRI). Human KLF4, ETV5, ARID4B, CDT1, ZMIZ1, JAZF1, and SUZ12 were sub-cloned into Gateway pENTR4 and then was recombined into the MSCV-DsRed-IRES-EGFP destination vector by Gateway cloning using LR clonase II enzyme. The TA Mut versions of the constructs (where the two phosphorylatable residues of the phosphodegron motif were replaced to alanine were the following: ETV5 (P41161-1; aa135 and 139), ZMIZ1 (Uniprot ID:Q9ULJ6-1; aa517 and 521), ARID4B (Uniprot ID: Q4LE39-1; aa1022 and 1026), JAZF1 (Uniprot ID: Q86VZ6-1; aa109 and 113), CDT1 (Uniprot ID: Q9H211-1; aa and 406), and KLF4 (Uniprot ID: O43474-3; aa394 and 398). JAZF1, SUZ12, and JAZF1-SUZ12 were sub-cloned into the pcDNA 3.1 (+)-cMyc (Invitrogen) vector using restriction enzymes (BamHI, KpnI, or NotI). The JAZF1-SUZ12 fusion was created using overlapping extension PCR.

### 4.3. Cell Culture

HEK293T cells (purchased from ATCC, CRL-3216) were cultured in Dulbecco’s Modified Eagle’s Medium (DMEM, Gibco, Waltham, MA, USA) containing 10% FBS (Gibco, Cat# 10500064), 100 µg/mL Pen/strep (Sigma, P0781) and Amphotericin B (Thermo Fisher, 15290026) at 37 °C in an atmosphere of 5% CO_2_. Approximately 90,000 cells were plated in 24-well Greiner bio-one plates (#662-160). The next day cells were transfected with Lipofectamine 3000 reagent (L3000001, Invitrogen, Waltham, MA, USA) using the Opti-MEM reduced serum media (Gibco, 31985062). The transfection media was changed with complete DMEM after 4–6 h of transfection. Cells in a 24-well plate were transfected with 500 ng of plasmid DNA in total, for co-transfection each plasmid construct was used in 250 ng amount. For western blots, cells were washed in ice-cold PBS and harvested in SDS loading buffer and then subjected to SDS-PAGE. The samples were then transferred to nitrocellulose membrane. Primary antibodies were the following: Anti-V5 (Invitrogen; 1:5000 dilution), anti-FLAG (Sigma #F1804; 1:5000 dilution), anti-cMyc (#2276, Cell signaling, Danvers, MA, USA) 1:1000 dilution) and anti-tubulin (Sigma #T6199; 1:50,000 dilution). Secondary antibodies (IRDye^®^ 800CW Goat anti-Mouse IgG, LiCOR #926-32210 and IRDye^®^ 680CW Goat anti-Mouse IgG, LiCOR #926-68070; 1:5000 dilution) were used according to the recommendation of the manufacturer and blots were analyzed by Odyssey CLx imaging system (Li-Cor). Kinase inhibitors were used in 2 µM concentration added 6 h before protein level measurements: GSK3—CHR99021 (#S2924, Selleckchem, Munich, Germany), ERK1/2—SCH772984 (Selleckchem, #S7101), p38—SB202190 (Sigma-Aldrich, #S7067), and JNK—JNK-IN-8 (Selleckchem, #S4901). MG132 proteosomal inhibitor (Merck, #474790) was used in 10 µM concentration and cycloheximide (# 239763, Calbiochem, San Diego, CA, USA) was used in 100 µg/mL concentration.

### 4.4. Protein Degradation Assay

HEK293T cells were washed with 500 μL ice-cold PBS twice and gently collected using a pipette 48 h after transfection. The suspension was centrifuged at 500× *g* for 10 min at 4 °C. After centrifugation, the cell pellet was resuspended in 110 µL ice cold PBS and the fluorescence signal for EGFP (Excitation 485 nm; Emission 515 nm) and DsRed (Excitation 550 nm; Emission 580 nm) was measured in Corning 384-well microplate (#4514) using a plate reader (CytationTM 3, BioTek Instruments, Winooski, VT, USA).

### 4.5. Protein Production and Purification

Truncated Skp1 (1-149) and human FBXW7 (263-707) with an N-terminal hexa-histidine (His6) tag were cloned into a bicistronic plasmid (pMG950) and were co-expressed in *E. coli* Rosetta (DE3) strain. The Skp1-FBXW7 complex was purified using affinity, anion exchange, and gel filtration chromatography. The DNA expression plasmid was transformed into *E. coli* Rosetta (DE3) strain and grown in LB until OD600 = 0.5, protein expression was induced with 100 µM isopropyl-β-d-1-thigalatopyaranoside (IPTG) overnight at 18 °C. The cell pellet was lysed, and the recombinant protein was purified using Ni-NTA agarose beads. Before further purification the His6-tag was removed by using the TEV protease and then the protein complex was further purified using anion exchange chromatography twice, first with a Resource Q and then with a Mono Q column (pH = 8). Finally, the complex was subjected to size-exclusion chromatography in 20 mM Tris, pH = 8, 150 mM NaCl, and 1 mM DTT using a Superdex200 10/300 column (GE Healthcare, Chicago, IL, USA). The eluted fractions were pooled together and were supplemented with 10% glycerol and 2 mM TCEP before concentration and storage.

### 4.6. Protein–Peptide Binding Assay

Fluoresce polarization (FP) based protein–peptide binding affinity measurements were carried out using 50 nM carboxyfluorescein (CF) labeled CCNE peptide (CF-PSGLLpTPPQpSGKKQ) and data were fit to a direct or a competitive binding equation using OriginPro 2018 as described earlier [23]. For the direct measurements the concentration of Skp1-FBXW7 complex was increased in a titration experiment and then the binding affinity (Ki) of the unlabeled peptide was determined by titrating the Skp1-FBXW7: labelled peptide complex sample, at a concentration corresponding to ~60–80% complex formation, with increasing amounts of unlabeled peptide. All the FP measurements were performed in 20 mM Tris (pH 8.0), 100 mM NaCl, 0.05% Brij35P, and 2 mM TCEP using a Cytation^TM^ 3 (BioTek Instruments) plate reader (384-well plates in 20 µL binding reaction volume). Peptides were chemically synthesized and were ordered from GenScript.

## Figures and Tables

**Figure 1 ijms-23-03320-f001:**
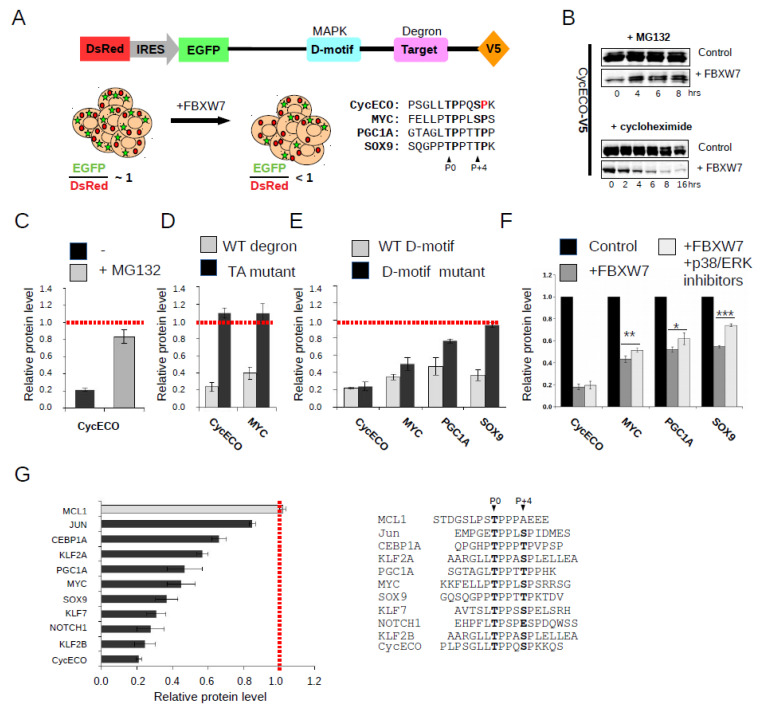
Design and validation of the protein degradation assay used for FBXW7 phosphodegron discovery. (**A**) The assay concept is based on elevated degradation of an EGFP-fused degradation probe upon heterologous FBXW7 expression. The probe contains a MAPK binding D(ocking)-motif (D-motif) to allow selective kinase recruitment and the putative FBXW7 binding phosphodegron (Target). EGFP fluorescence signal is compared to the DsRed signal, and similar expression of DsRed and the EGFP-probe is ensured by the internal ribosome entry site (IRES) between these two gene products. If the phosphodegron motif is functional, the EGFP/DsRed fluorescence signal ratio is expected to drop from 1 to a lower value. The target motifs that were used for the validation of the assay are shown with their kinase phosphorylation target sites (S/TP) highlighted in bold. CycEco (cyclinE C-terminal optimized degron) is a modified motif in which a glycine residue was modified to proline, shown in red, to make it compatible for MAPK-mediated double-phosphorylation. (**B**) Steady-state CycECO-V5 protein levels after inhibiting the proteosome (+MG132) or translation (+cycloheximide). Panels show the results of MG132 (10 µM) or cycloheximide treatment (100 µg/mL) using western blots. HEK293T cells were transfected with empty plasmid (control) or with the FBXW7α expression plasmid (+FBXW7) and the level of the CycECO chimera was monitored using an anti-V5 antibody after drug treatment in time. (**C**) CycEco-EGFP probe degradation is blocked by a proteosome inhibitor (MG132). (**D**) Intact phosphorylation target sites are required for probe degradation. WT: wild type; TA mutant: the phosphorylatable residues at position 0 (P0) or P + 4 are replaced by alanines (**A**). (**E**) Phosphodegron mediated degradation depends on MAPK recruitment for MYC, PGC1A, and SOX9 derived motifs, but CycECO mediated degradation was unaffected. (**F**) Kinase inhibitor treatment confirmed that, apart from CycECO, MAP kinases (ERK1/2 and p38) contribute to probe degradation. HEK293T cells (Control) were transfected with the FBXW7α expression plasmid and were untreated (+FBXW7) or incubated with a mix of ERK1/2- (SCH772984; 2 µM) and p38-specific (SB202190; 2 µM) inhibitors (+FBXW7 + p38/ERK inhibitors). (**G**) Performance of known FBXW7 binding motifs. In addition to CycECO, 9 out of 10 motifs were tested positive, where +FBXW7/control protein level ratio was lower than 1 (shown in black). Panel C-F show relative protein levels normalized to the EGFP/DsRed ratio determined for the degron probe when only an empty vector without FBXW7 cDNA was transfected (control). The basal EGFP/DsRed ratio was about 1 for all constructs tested (shown with a red line). Error bars show SD based on three independent experiments. Statistical significance was calculated based on two-tailed paired Student’s *t*-test (*: *p* < 0.05; **: *p* < 0.01; ***: *p* < 0.001).

**Figure 2 ijms-23-03320-f002:**
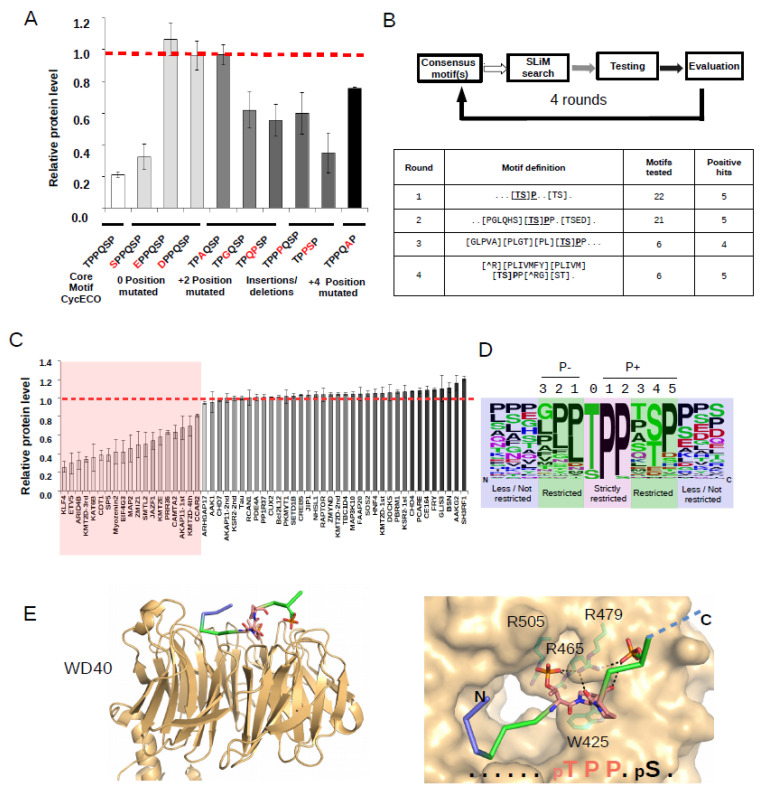
Refinement of the FBXW7 phosphodegron consensus. (**A**) Mutational analysis with the CycECO degron. The relative protein level of the CycECO probe was monitored in HEK293T cells (where the EGFP/DsRed ratio measured in FBXW7 plasmid transfected cells was normalized to the EGFP/DsRed ratio of control cells, where the latter were not transfected by the FBXW7 expression plasmid). Amino acid replacements/changes compared to the TPPQSP core motif are shown in red. (**B**) The scheme of the hierarchic strategy that coupled sequence pattern-based motif identification from the disordered part of the human proteome with experimental testing of chosen motifs in the ratiometric EGFP/DsRed degradation assay. The underlined residues denote the P0 position in the degron motif. (**C**) Experimental testing of putative motifs identified 19 novel phosphodegrons (ordered from strong to weak). (**D**) Sequence logo created based on all formerly known positive and the newly found 19 human positive phosphodegrons (see Appendix A). (**E**) Position of strictly restricted, restricted, and less/not restricted phosphodegron regions in the FBXW7(WD40)-CCNE peptide complex (PDB ID: 2OVQ) [12]. The crystallographic model of the WD40 domain bound to the CCNE peptide (LPSGLLpTPPQpSG) is shown from the side on the left panel, while the panel on the right shows a zoomed-in view of the structure from the top; peptide regions are colored as on Panel D; N denotes N-terminus and C denotes C-terminus. The core TPP sequence of the CCNE peptide is shown in stick representation and is colored in salmon. Notice how the proline in P + 1 snugly fits between W425 and R465 on this otherwise shallow protein–protein interaction surface. In addition, P + 2 may be important in positioning the phosphorylated serine (pS) by constraining the peptide backbone. The WD40 domain is shown with surface representation where contact residues (R465, R479, R505, and W425 are shown in stick representation and are colored in cyan). H-bond contacts between peptide atoms and protein residues are shown with black dashed lines.

**Figure 3 ijms-23-03320-f003:**
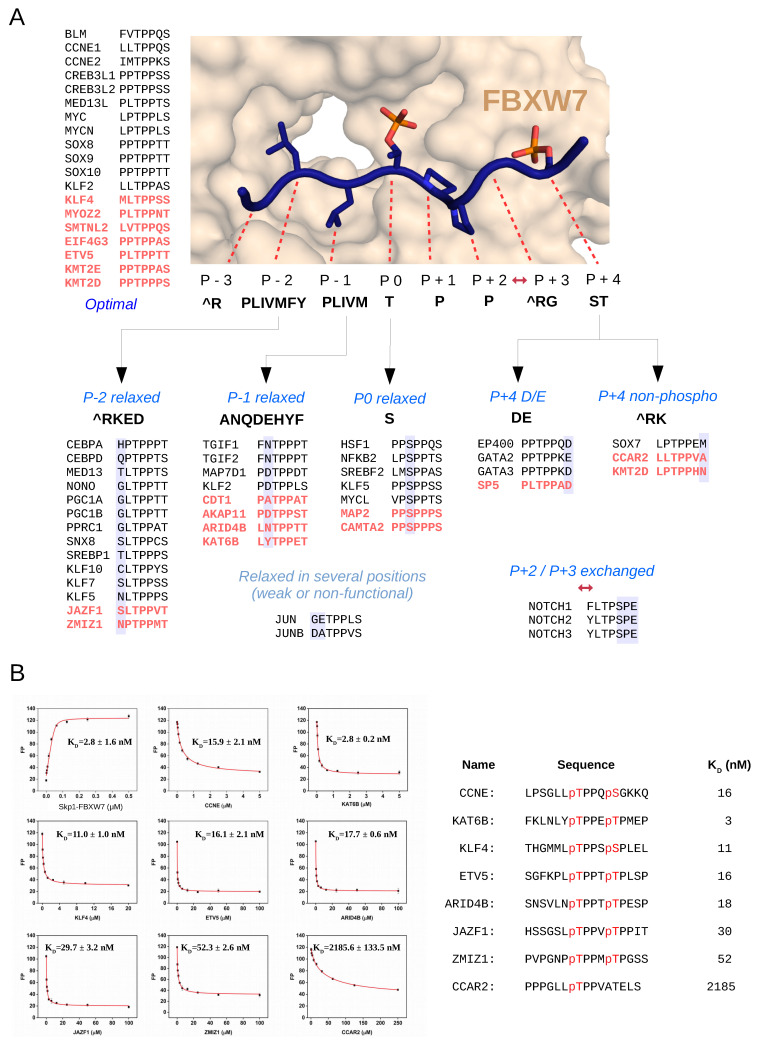
Newly identified phosphodegrons bind to FBXW7 with nanomolar affinity. (**A**) The scheme of the position–relaxation model describing FBXW7 binding phosphodegron motifs. The structural panel shows the crystal structure of the FBXW7(WD40)-CCNE peptide complex (PDB ID: 2OVQ) highlighting the different motif positions (from P − 3 to P + 4). The optimal sequence pattern, shown in the middle, was defined based on the analysis shown on Figure 2. Motifs with a relaxed position, namely with a sub-optimal amino acid, are shown below. The motif sequence newly identified in this study are shown in red, while known motifs are shown in black; motifs are listed according to the different position–relaxation model classes (see Appendix A). Note that motifs that deviate from the optimal pattern in more than one position may still bind, albeit with weaker affinity (e.g., JUN and JUNB). (**B**) Results of quantitative protein–peptide binding assays based on measuring fluorescence polarization. The first panel shows the result of the direct titration experiment with the fluorescently labeled CCNE peptide and increasing amounts of the Skp1-FBXW7 complex. Further panels show the results of competitive binding experiments where the binding of the fluorescently labeled CCNE peptide was competed off by different unlabeled peptides shown on the right (pT: phosphor-threonine and pS: phosphor-serine).

**Figure 4 ijms-23-03320-f004:**
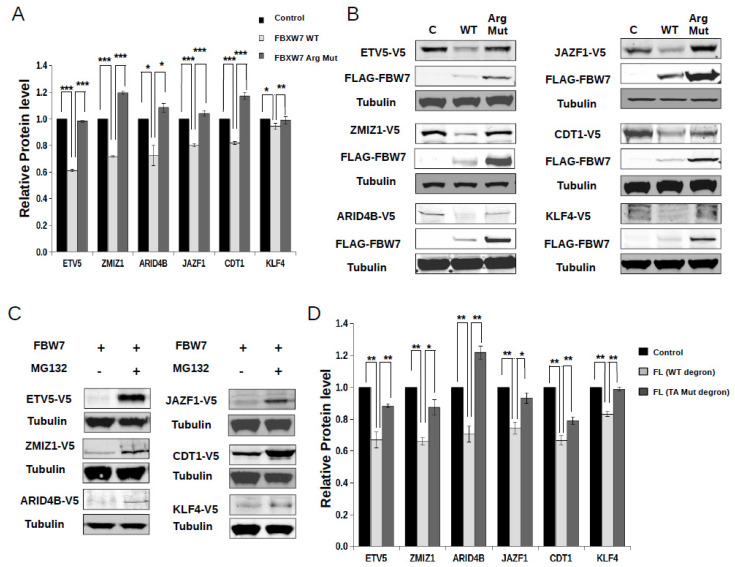
Phosphodegrons in full-length proteins. (**A**) Intact phosphodegron-FXWB7 binding is essential for the regulation of full-length protein levels. The level of full-length proteins containing an FBXW7 interacting phosphodegron were analyzed by the EGFP/DsRed ratiometric assay in HEK293T cells 48 h after transfection with empty plasmid (Control), or with the expression plasmid for wild-type (WT) or mutated FBXW7 (Arg Mut: three arginine amino acids mutated in the WD40 domain important for phosphodegron-FBXW7 binding). (**B**) Similar experiment to Panel A, but protein levels were monitored by western blots. Each full-length protein construct was expressed with a V5-tag and the heterologously expressed FBXW7 constructs had a FLAG-tag. Anti-tubulin was used as the load control. (**C**) FBXW7-mediated degradation of the full-length protein constructs are inhibited by the MG132 proteosomal inhibitor. Western blot analysis was conducted similarly to Panel B. (**D**) Phosphorylation of the phosphodegron motifs in full-length protein constructs is essential for their FBXW7 mediated degradation. The experiment was carried out similarly as shown on Panel A. Control: co-transfection with intact full-length protein constructs and empty FBXW7 expression vector; FL (WT degron): co-transfection with intact phosphodegron motif containing full-length constructs and wild-type FBXW7; FL (TA Mut degron): co-transfection with mutated phosphodegron motif (two phosphorylatable residues replaced with alanines) containing full-length constructs and wild-type FBXW7. Relative protein levels show the EGFP/DsRed ratio for the different full-length constructs. Error bars show SD based on three independent experiments. Statistical significance was calculated based on two-tailed, paired Student’s *t*-test (*: *p* < 0.05; **: *p* < 0.01; ***: *p* < 0.001).

**Figure 5 ijms-23-03320-f005:**
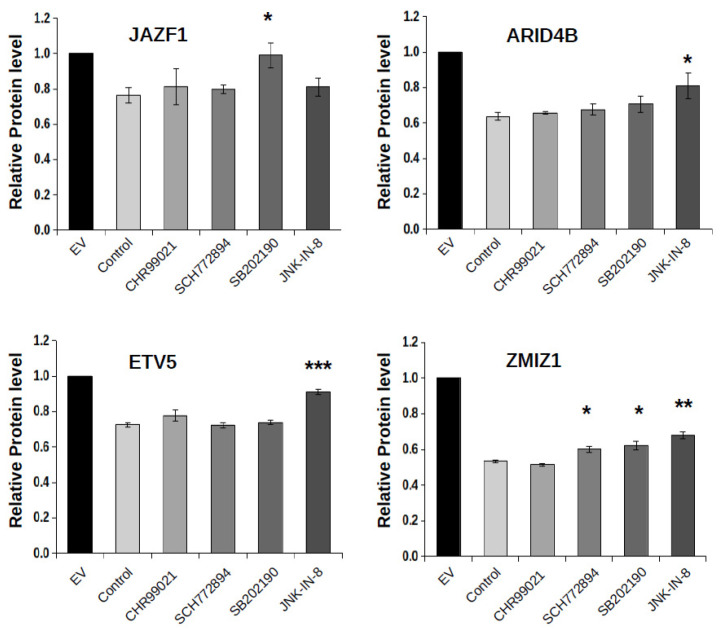
Effects of kinase inhibitors on FBXW7-mediated protein degradation. The ratiometric DsRed/EGFP protein level assay was used to monitor the effects of kinase inhibitor treatments on the level of four different phosphodegron containing full-length proteins in HEK293T cells: JAZF1, ARID4B, ETV5, and ZMIZ1. Inhibitors (CHR99021, GSK3 inhibitor; SCH772984, ERK1/2 inhibitor; SB202190, p38α/β inhibitor; JNK-IN-8, JNK1-3 inhibitor) were added in 2 µM concentration 6 h before DsRed/EGFP measurements. EV: cells transfected with empty vector; Control: cells transfected with the wild-type FBXW7 expression plasmid. Relative protein levels show the EGFP/DsRed ratio for the different full-length constructs. Error bars show SD based on three independent experiments. Statistical significance was calculated based on two-tailed, paired Student’s *t*-test (*: *p* < 0.05; **: *p* < 0.01; ***: *p* < 0.001).

**Figure 6 ijms-23-03320-f006:**
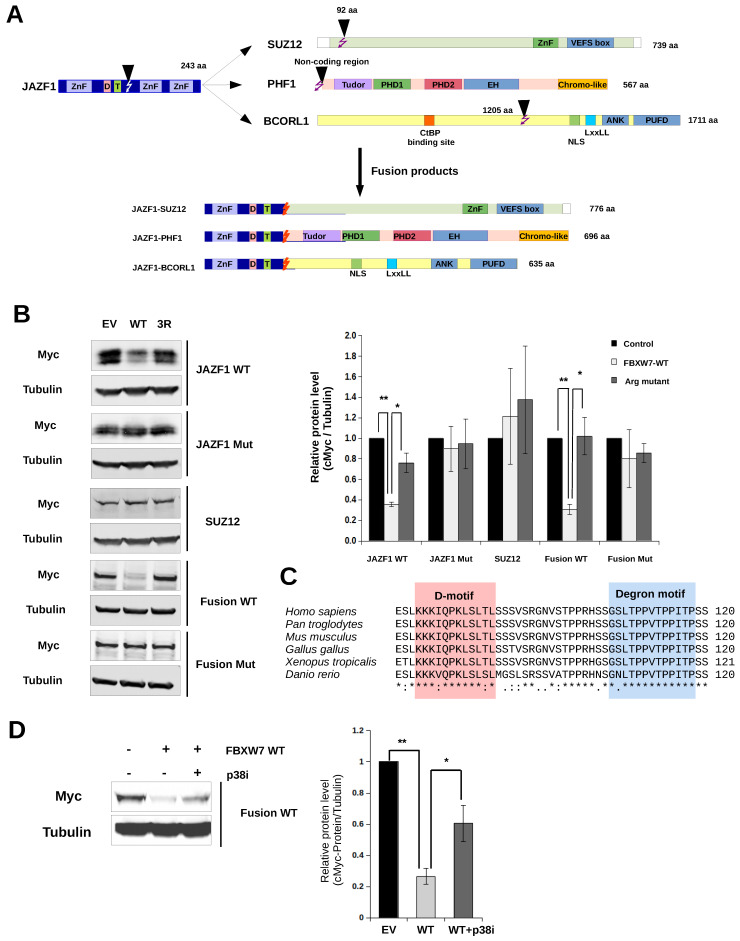
Regulation of JAZF1-SUZ12 fusion protein level by p38-FBXW7 partnership. (**A**) Scheme of JAZF1 fusions known to be associated with endometrial stromal sarcoma. (**B**) Western blot analysis of HEK293T cells transfected with Myc-tagged JAZF1 wild-type (WT), JAZF1 Mut (phosphodegron mutant), SUZ12, JAZF1-SUZ12 “wild-type” fusion (Fusion WT), and JAZF1-SUZ12 fusion with phosphodegron mutations (Fusion Mut). EV: cells co-transfected with empty vector; WT: cells co-transfected with wild-type FBXW7; Arg Mut: cells co-transfected with FBXW7 mutated at three arginine positions to eliminate WD40 domain-phosphodegron binding. The bar graph on the right shows the quantification of the western blot (WB) signal calculated based on three independent experiments. Relative protein level was calculated based on the ratio of Myc-protein and Tubulin WB signals and this value was normalized to that of cells transfected with the empty vector (EV). Error bars show SD based on three independent experiments. Statistical significance was calculated based on two-tailed, paired Student’s *t*-test (*: *p* < 0.05; **: *p* < 0.01). (**C**) Multiple sequence alignment of JAZF1 regions from human, mouse, chicken, a frog (*Xenopus tropicalis*), and a fish (*Danio rerio*). The FBXW7 binding degron region and the N-terminally located MAPK-binding D-motif is boxed. (**D**) Western blot analysis of cells co-transfected with JAZF1-SUZ12 fusion (Fusion WT) and FBXW7 in the absence or presence of a p38-specific inhibitor (2 µM; p38i: SB202190). A representative anti-Myc and anti-Tubulin WB is shown on the left (-FBXW7 refers to EV where cells were not transfected with an FBXW7 containing expression vector, only with an empty vector). The right panel shows that FBXW7 co-expression greatly decreases the protein level of Fusion WT, which is counteracted by the kinase inhibitor. Error bars show SD based on three independent experiments. Statistical significance was calculated based on two-tailed, paired Student’s *t*-test (*: *p* < 0.05; **: *p* < 0.01).

**Figure 7 ijms-23-03320-f007:**
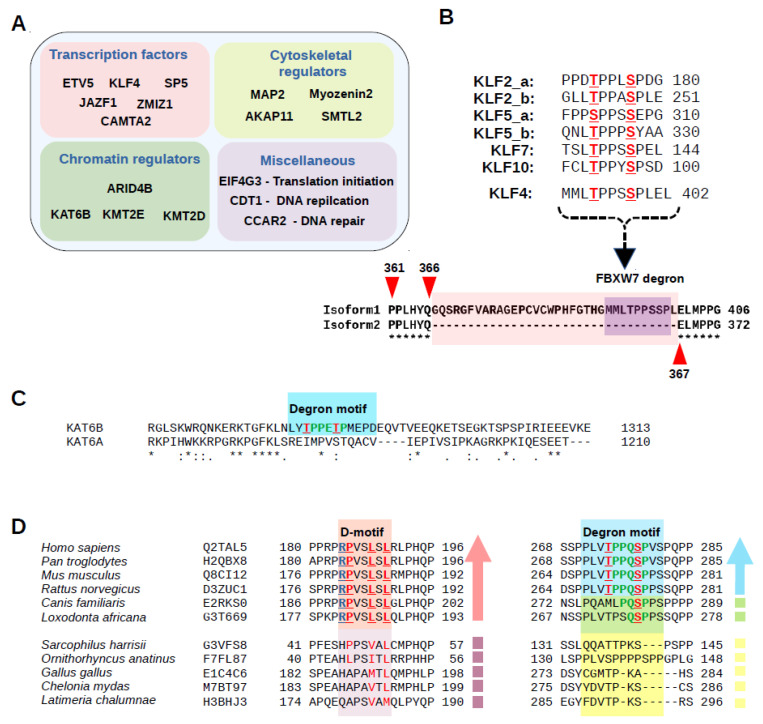
Sequence analysis of proteins containing experimentally validated phosphodegrons. (**A**) New experimentally tested FBXW7 phosphodegron motifs located in the disordered region of the proteins are classified into different categories based on their functionality: transcription factors, chromatin regulators, cytoskeletal regulation, and some other proteins with miscellaneous functions. (**B**) Sequence alignment showing phosphodegron motifs of the 5 KLF family members. The most studied isoform of KLF4 lacks a small region (34 aa) including the FBXW7 phosphodegron motif. (**C**) Alignment of KAT6B and KAT6A showing that evolutionary conserved FBXW7 phosphodegron motif is present in KAT6B but is absent in its paralog KAT6A (* indicates conserved positions). (**D**) Sequence analysis of the SMTL2 protein hints on the co-evolution of a MAPK binding docking motif and the FBXW7 binding phosphodegron motif.

## Data Availability

Not applicable.

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
