# Peer review of "Systematic Discovery of FBXW7-Binding Phosphodegrons Highlights Mitogen-Activated Protein Kinases as Important Regulators of Intracellular Protein Levels"

_ijms, 2022, doi:10.3390/ijms23063320_

Round 1
Reviewer 1 Report
This manuscript by Singh and co-workers defines a so-called “phosphodegron” sequence motif targeted by the E3 ubiquitin ligase component FBXW7, and important mediator of signal-induced protein degradation and an established tumor suppressor gene. To do so they leverage a new system for reading out protein stability in living cells in which they use a reporter construct linking a kinase recruitment motif to candidate phosphodegron sequences. Through iterative cycles of testing candidate sequences, motif refinement, and database searching, they establish a consensus sequence highlighting key features either required or promoting protein phosphorylation and/or degradation. They subsequently identify and validate a set of full length proteins harboring functional phosphodegrons, and demonstrate that their FBXW7-mediated degradation is at least partly blocked by inhibiting one of the major MAP kinase families (ERK, p38 or JNK). Finally, they show that a recurrent fusion protein found in endometrial tumors is degraded by FBXW7 in a phosphorylation-dependent manner.
This is an interesting study that has important implications for our discovery of new FBXW7 substrates, and should be of strong interest to those in the field. I have only a few minor comments:
- In the introduction, it should be stated that GSK3beta itself recognizes substrates “primed” by prior phosphorylation, so that phosphodegrons in general would be formed by the action of a priming kinase and then GSK3beta.
- In Fig 1F, the effects of p38/ERK inhibitors on reporter degradation are surprisingly modest, suggesting other kinases may be active or that experiments were not done under conditions of high MAPK activity. Is it also possible that mono C-terminal phosphorylation is stabilizing (as it is for Myc protein). The authors should acknowledge these possibilities.
- Similarly, in Fig 5, in almost all cases where there was an effect of MAPK inhibitors on protein levels, the effect was only partial, suggesting other kinases may contribute. For example multiple MAPKs could converge on the same site.
- In Fig 6, “cMyc” should be replaced with “Myc” throughout to avoid confusing the epitope tag with authentic c-Myc protein.
- On p.17 the authors state “…our study is the first that shows that natural JAZF1 level is regulated by the FBXW7 ubiquitin ligase in a MAP kinase dependent fashion.” This was only shown in the context of an artificial construct, not in the context of the “natural” or endogenous protein, and the authors should acknowledge that limitation.
Author Response
1) The role of GSK3 is explicitly mentioned now in the Introduction as suggested (see Line 83 in the revised version).
2) Regarding the description of the data shown on Figure 1F, the following sentence was added to the text (see Line 155).
"The inhibition exerted by these MAPK-specific inhibitors on FBXW7 mediated degradation was only partial. This may be because the degradation experiments were done under low MAPK activity conditions, other kinases may also be active on the degradation probes, or mono- or double-phosphorylation of the phosphodegron motif may have somewhat distinct effects."
3) Regarding the description of the data shown on Figure 5, the following text was added (see Line 411):
"The effect of MAPK inhibitors were modest in most cases, possibly because other kinases are also active on these phosphodegrons. Notwithstanding, the data suggest ..."
4) "cMyc" was changed to "Myc" on Figure 6 as suggested.
5) We changed the warding of this sentence to (see Line 585):
"Based on our experiments using artificial constructs, to our knowledge, our study is the first that suggests that natural JAZF1 level could be regulated by the FBXW7 ubiquitin ligase in a MAP kinase dependent fashion."
Reviewer 2 Report
In this manuscript, Neha Singh and collaborators identified phosphodegrons located in several proteins, including those involved in chromatin regulation (ARID4B, KMT2E, KMT2D, 16 KAT6B) or cytoskeletal regulation (MAP2, Myozenin-2, SMTL2, AKAP11), and some other proteins with miscellaneous functions (EIF4G3, CDT1, CCAR2). They showed that the protein level of full-length ARID4B, ETV5, JAZF1, and ZMIZ1 are affected by different MAPKs inhibitors, suggesting that MAPK and FBXW7 partnership plays an important cellular role by directly affecting the level of key regulatory proteins. In addition, they found that the p38α-controlled phosphodegron in JAZF1 may be responsible for the pathological regulation of the cancer-related JAZF1-SUZ12 fusion construct implicated in endometrial stromal sarcoma.
Overall, the manuscript is of interest. However, English editing revision is recommended, and the following comments/questions should be addressed:
Fig 5: Specific inhibitors for ERK1/2, p38 and JNKs have been used. What about ERK5? Did the authors test the effects of ERK5 inhibitors on FBXW7-mediated protein level control?
References: some information are missing in ref #41, 56
Author Response
We provided the missing information regarding ref#41 and 56. ERK5 inhibitor was not used, beacuse we found earlier that ERK5 expression is low and we posited there is not enough basal ERK5 activity in HEK293T cells to be able to judge any impact of this MAPK on FBXW7 mediated degradation. We carefully edited the text, fixed some typos and improved the English as suggested.